# Continuous Material Deposition on Filaments in Fused Deposition Modeling

**DOI:** 10.3390/polym16202904

**Published:** 2024-10-15

**Authors:** Guy Naim, Shlomo Magdassi, Daniel Mandler

**Affiliations:** Institute of Chemistry, The Hebrew University of Jerusalem, Jerusalem 9190401, Israel; guy.naim3@mail.huji.ac.il (G.N.); shlomo.magdassi@mail.huji.ac.il (S.M.)

**Keywords:** 3D printing, fused deposition modeling, preprinting, filament treatment, coating, continuous material deposition on filaments (CMDF), PLA, ZnO, rhodamine B, ciprofloxacin

## Abstract

A novel approach, i.e., Continuous Material Deposition on Filaments (CMDF), for the incorporation of active materials within 3D-printed structures is presented. It is based on passing a filament through a solution in which the active material is dissolved together with the polymer from which the filament is made. This enables the fabrication of a variety of functional 3D-printed objects by fused deposition modeling (FDM) using commercial filaments without post-treatment processes. This generic approach has been demonstrated in objects using three different types of materials, Rhodamine B, ZnO nanoparticles (NPs), and Ciprofloxacin (Cip). The functionality of these objects is demonstrated through strong antibacterial activity in ZnO NPs and the controlled release of the antibiotic Cip. CMDF does not alter the mechanical properties of FDM-printed structures, can be applied with any type of FDM printer, and is, therefore, expected to have applications in a wide variety of fields.

## 1. Introduction

Among a large number of 3D-printing techniques, fused deposition modeling (FDM) is probably the most common. Therefore, introducing new functional filaments for FDM opens additional avenues for various applications. In recent years, FDM printing has gained a lot of interest in improving the process and achieving printed objects with improved properties. This includes incorporating various materials within the filaments [1,2], the study of new polymers as filaments [3,4,5], the insertion of fibers directly into the nozzle during printing [6,7], and localized re-heating of the deposited filament through radiation or heat [8,9,10,11,12,13], to name a few. Modifications to manufactured FDM-printed objects can be carried out during filament fabrication (premanufacturing treatment) directly to a manufactured filament before printing (preprinting treatment), or after performing the printing process (post-printing treatment). Premanufacturing treatments are often performed by applying some sort of modification or addition to the polymer pellets before the production of the filament. Common examples include the addition of additive materials to the process or treating the polymer pellets before melting them (i.e., by grafting or oil casting) [14]. In a fairly similar manner, such treatments can be carried out during the printing process by using nozzle impregnation to insert foreign material into the print or by melting a polymeric mixture directly into the nozzle, skipping filament fabrication [15,16,17]. Post-printing is far more common than preprinting processes and usually involves surface treatments, such as dip coating [18], spray coating [19,20], and electroless plating [21,22], among others. Two key approaches are used for preprinting treatments. The first, and seemingly more common, utilizes the swelling nature of certain polymers immersed in organic solvents as a method of inserting additives into the filament [23,24,25], while the other is based on the adsorption of additive materials as a coating [26,27,28].

Preprinting treatment holds certain advantages over post-printing, which include improved uniformity of the added material in the printed samples, a stronger binding, and a ready-to-use object upon printing. An interesting comparison between post- and preprinting treatments was reported by Farto-Vaamonde et al., who incorporated antibiotics and carvacrol into polylactic acid (PLA) [29,30]. They found a strong effect on the release rates of the embedded materials, where preprinting showed a slower delayed release.

Despite the clear advantages of preprinting treatment over post-printing, it does bear certain additional challenges, such as the need to ensure a uniform layer of coating before and after the printing process, the high temperature encountered during the FDM printing, and the unfit geometry of the filament to certain coating processes, such as spraying and spin coating. Therefore, most studies turn to post-printing or premanufacturing of the filament as a method of sample modification. Among the studies that are focused on preprinting treatments, the majority utilize the swelling nature of the polymers to include active materials. Only a handful of reports applied a preprinting modification through the adsorption of additive materials dissolved in a solution containing the same polymer as the filament. Francis et al. applied this method to acrylonitrile–butadiene–styrene filaments, carrying montmorillonite-based nanocomposites into the forming layer, thus improving both the relative permittivity and mechanical properties of the printed samples [31]. Wang et al. wrapped hydroxyapatite nanoparticles (NPs) with PLA before attaching them to PLA filaments as a coating, attempting to better meet the physical requirements of load-bearing bone implants [32]. This preprinting approach is appealing and adaptable, as it results in a uniformity of the active material as long as it can be attracted to the dissolved polymer and allows the functionalization of filaments without changing their printability.

Despite that, certain problems and limitations need to be overcome. Often, the preprinting treatment of filaments is carried out by cutting them prior to treatment and remelting them together afterward [29,30,31]. This is a challenging, as should the seams not be perfectly aligned, the printer might clog during printing and the amount of filament drawn by the motors might be inconsistent. Certain studies overcome this problem by treating the entire filament [28,32]. However, in those cases, a large volume of the coating solution was necessary and the filaments needed to be separated when the coating material caused bundles of filaments to form. Sweeney et al. suggested a continuous system, termed continuous bath coating, where the filament was drawn through a coating solution, solving the challenges associated with cutting and rebinding or bundling of the filaments [33,34,35]. One major flaw of the continuous bath coating approach stems from the friction acting on the filament during the process, resulting in an uneven coating. Another challenge of the continuous bath coating approach, in addition to the other methods, arises from the volatility (and toxicity) of the organic solvents used, which gradually change the concentration of the additives and the dissolved polymer, thus affecting the coating layer of the filament.

In view of the above, an improved preprinting treatment system for filaments, where the filament does not come into contact with the bath walls during the process and the evaporation of the organic solvent is slowed, could produce better results. Our approach to the preprinting treatment of FDM filaments is presented schematically in Figure 1. The filament is immersed within a half-circular tube containing an aqueous salt solution, a PLA-saturated dichloromethane (DCM)/tetrahydrofuran (THF) (1:1 *v*/*v*), and a covering end layer of distilled water. The potential friction from the surface of the bath is completely avoided by pulling the filament in one direction only during the coating and the initial drying steps. Furthermore, the evaporation of the organic phase is greatly diminished by covering the DCM/THF with a layer of distilled water. The importance of using both a salt-saturated solution and distilled water stems from the difference in density; the first is denser than the hydrophobic phase, while the second is lighter.

Using this novel design, we have successfully coated a PLA filament with either Rhodamine B as a model material for optical and fluorescence microscopy, antibacterial ZnO NPs as a model for the entrapment of NPs, and the antibiotic Ciprofloxacin (Cip), to better understand the release kinetics of organic additives. Further details about the materials and how they were studied can be found in following sections. We have found that the filaments were evenly and uniformly coated, the coatings had survived the printing process, and the coated materials showed high activity in the printed structures, despite the small amount of material used per filament length. 

## 2. Materials and Methods

### 2.1. Materials

PLA filaments with a diameter of 1.75 mm were purchased from Esun (Chi Minh, Vietnam) and PRUSA (Prague, Czech Republic) and were used interchangeably. Most of the chemicals were purchased from Sigma-Aldrich-Merck, namely Rhodamine B, potassium nitrate ≥99.0%, ZnO NPs (nanopowder, <100 nm particle size), sodium phosphate monobasic (AR), sodium phosphate dibasic (AR), hydrochloric acid 32%, and Zincon monosodium salt. Edible NaCl was used. BACTO agar, BACTO yeast extract, and BACTO tryptone were from Thermo Fisher Scientific (Bleiswijk, The Netherlands). Solvents, including dichloromethane (DCM), tetrahydrofuran (THF), and acetonitrile (ACN), were purchased from BioLab Ltd. (Jerusalem, Israel). Ciprofloxacin ≥ 99% was provided by TZAMAL-D-CHEM (Petach Tikva, Israel). The chemicals were used as received. The bacteria *Escherichia coli* (*E. coli*) and Staphylococcus aureus (*S. aureus*) were cultivated in-house. 

### 2.2. Instruments

A Prusa i3 MK3S + 3D printer and the slicer software used, Pruseslicer 2.5.2, were ordered and downloaded from PRUSA (Prague, Czech Republic). An orbital shaker incubator (MRC, Holon, Israel) was used for cultivating the bacteria and simulating body-mimicking conditions, and a 3150 Tuttnaur autoclave was used for sterilization. The mechanical properties were measured with an Instron 3345 (Instron, Norwood, MA, USA). Images of the samples were taken using a fluorescence microscope (Axio Scope.A1, Carl Zeiss, Germany) with a 543 nm excitation and 565 nm emission filter, optical microscopes from Carl Zeiss (Jena, Germany), and an Olympus BX60 (Tokyo, Japan). UV-Vis spectroscopy measurements were conducted using a UV-1900i UV-VIS Shimadzu spectrophotometer (Nakagyō-ku, Kyoto, Japan).

### 2.3. Methods

#### 2.3.1. Coating Solution Preparation

First, 10 mL of DCM and 10 mL of THF are stirred at 600 rpm at 38 °C in a closed glass vial. Clear PLA filament is added and dissolved under continuous stirring to achieve a concentration of up to 20 mg/mL. After 30 min, when no solid PLA strands are visible in the solution, the active material, e.g., Rhodamine B, is added. The solution is stirred for an additional 5 min after the solution is visibly homogenous. 

#### 2.3.2. Coating Setup

The semicircle glass tube (shown schematically in Figure 1 and in a photo in Figure 2) in the coating system was constructed with glassware used for organic synthesis, i.e., two vacuum adapters and a thermometer adapter. A new strand of filament is guided inside the tube using a small, motorized wheel (Figure 2A) moving the filament at a speed of 7.5 cm∙min^−1^. First, the filament moves through around 40 mL of saturated NaCl and KNO_3_ solution (Figure 2A-Sec. 1). The volume was dictated by the size of the bath to ensure that the filament would not come into contact with the sides of the bath. This is the only section where the filament comes into contact with the bath walls. Next, the filament is pulled through the 20 mL hydrophobic DCM/THF phase (Figure 2A-Sec. 2). The dissolved PLA in this solution precipitates onto the surface of the passing filament, trapping the active materials in the solution with it. Next, the coated filament goes through a thin layer of around 0.5 mL of distilled water (Figure 2A-Sec. 3), which cleans the filament from the hydrophobic phase and serves as a barrier between the organic phase and the environment, preventing direct evaporation. The total distance the filament travels inside the liquid phases is approximately 35 cm. The organic phase makes up around 8 cm of the total length. Between the bath and the motor collecting the product (Figure 2A-Sec. 5), the coating is dried upon exposure to air (Figure 2A-Sec. 4). Keeping a distance greater than 20 cm between the distilled water phase and the collection motor allows the filament to dry sufficiently for it to be rolled and collected. The color of the filament changes according to the added material (Figure 2B). The filament is then dried for an additional 4 h at 50–60 °C. Thermally sensitive materials, such as Cip, were not dried using a second step.

#### 2.3.3. FDM Printing

The desired objects were sliced using Pruseslicer 2.5.2 software and printed by a Mk3S + Prusa FDM printer. Most of the printed objects were disks (10 mm diameter, 3 mm height) for antibacterial measurement and kinetic release experiments. In addition, dogbone shapes for mechanical testing and aligned rectangular cuboids (where all layers were printed facing the same orientation instead of the 90° default) were used for microscopy imaging. The printing parameters are described in Appendix A.

#### 2.3.4. Antibacterial Tests

Antibacterial activity was tested as described previously, with a method based on optical density (*O.D.*) measurements [36,37]. The viability percentage was estimated according to the following Equation (1): (1)$$O.D.\;\left(\%\right)=\frac{O.D._{s}}{O.D._{c}}\times 100\%$$
where *O.D._s_* and *O.D._c_* are the average optical densities of the samples and the control, respectively. The control experiments were performed for filaments coated with the same PLA coating solution but without the ZnO NPs, so that only the existence of the NPs is the variation between the two. Each assay was performed five times and the control group was tested for every individual plate.

#### 2.3.5. The Leaching of Zn(II) and the Presence of Zn(II) in Samples

The leaching of Zn(II) ions was determined by measuring the absorptance of the zinc–Zincon complex spectrophotometrically [38]. Printed samples with different concentrations of ZnO NPs were placed in 10 mL of sterilized phosphate-buffered saline (PBS) under shaking (120 rpm, 37 °C). The concentration of Zn(II) ions was measured every week for 9 weeks. To determine the total amount of Zn in the printed samples or in the treated filaments, a known mass was immersed in 3 mL of 5 M NaOH solution (to decompose the PLA) for 3 days (120 rpm, 37 °C). The pH of the solution was then acidified using 800 µL of HCl 32% to completely dissolve any ZnO NPs. After 2 days, the solution was diluted to a final volume of 10 mL with a borate buffer of 0.5 M and a pH of 9. A sample of 100 µL was further diluted with 850 µL of borate buffer and 50 µL of Zincon solution 1.6 mM and measured spectrophotometrically at 615 nm. The concentrations were determined by comparing the absorbance to that of a calibration curve obtained from known concentrations of ZnCl_2_.

#### 2.3.6. Tensile Testing

The Young’s modulus *(E)*, yield strength, ultimate strength, and max elongation were measured using dogbone-shaped samples printed in two orientations, XY and Z (Appendix A), to test the mechanical properties of the mesh and adhesion between layers. All measurements occurred under a tension speed of 5 mm/min at room temperature.

#### 2.3.7. Cip Release and Entrapment

The release of Cip from the samples immersed in 10 mL PBS under incubation (120 rpm, 37 °C) was followed periodically (up to 35 days) by measuring the absorptance at 271 nm. To determine the total amount of Cip in the printed samples, the samples were first dissolved in ACN, followed by the addition of 0.1 M HCl to cause PLA precipitation and obtain a PLA-free Cip solution (Appendix A).

## 3. Results and Discussion

The essence of CMDF involves passing a polymeric filament through a saturated solution of the filament material and the substance to be embedded. Specifically, we demonstrate the scope of this approach by coating a PLA filament with three materials, Rhodamine B, ZnO NPs, and Cip, followed by their use in a 3D printer. Through the inclusion of materials with different natures and their possible application, we hope to highlight the strengths of CMDF as a system with broad possibilities.

### 3.1. The Rhodamine B Coating

Rhodamine B was initially used as a model material for optical tracking and characterizing the coating, due to its intense color and fluorescence. The cross-section of the filament that passed through the coating solution containing 45 mM of Rhodamine B is shown in Figure 3. The optical images show a clear and homogenous coating of the filament with a thickness of approximately 45 μm (based on the cross-section image). Further imaging using fluorescence microscopy (Figure 4) shows that Rhodamine B coats the periphery of the filament with a thin and uniform layer, which maintains its integrity quite well upon printing. Notice (in Figure 3 and Figure 4) that the filament that is originally ca. 1.8 mm in diameter decreases to a structure that is approximately 0.4 mm × 0.35 mm due to its passage through a 0.4 mm nozzle and a layer height (determined by the software) of 0.35 mm. Hence, the microscope images confirm that the coating around the filament is maintained as a thin layer after printing, supporting its use in distinct locations in the printed samples. Keeping the coating as an outer brim enables control over only the surface properties, with no changes to the bulk of the material.

### 3.2. The ZnO NP Coating

Nanomaterials are particles with at least one dimension below 100 nm. The most studied of nanomaterials are NPs, benefiting medicine, energy harvesting and storage, and sensing. The inclusion of nanomaterials in 3D objects can present unique advantages. Metal oxide NPs, such as ZnO, are among the most promising of materials for medical implants because of their potent antimicrobial activity, including antibiotic-resistant bacteria [39]. ZnO NPs have a low toxicity and are FDA-approved for broad application as a highly efficient antibacterial and antifungal material [40]. Therefore, we evaluated the applicability of our system to ZnO NPs. The resulting printed objects should have a uniform coating around and along the filament and provide an antibacterial activity without affecting the mechanical properties of the PLA. Accordingly, we applied CMDF to coat the PLA filament by passing it through a dispersion of ZnO NPs, as described in Section 2.3.2. The amount of ZnO around the filament (Figure 5) and the printed samples (Figure 6) was analyzed by a spectrophotometric method based on the complexation by Zincon in samples obtained by dissolving the PLA in NaOH and adjusting the acidity (see Section 2.3.5).

The amount of ZnO NPs was evaluated along the filament length and as a function of the ZnO concentration in the coating dispersion. As seen in Figure 5, the amount of Zn in all sections within the first meter of the filament decreases until it reaches a constant value. We expect this trend to continue in long filaments, changing only after a noticeable reduction in the concentration of ZnO NPs in the coating solution. In other words, the uniformity of the coating between the first and third meter could be maintained in longer filaments. For example, 35 m of filament coating would cause a decrease of only 5% of ZnO NPs in the dispersion. This indicates that CMDF holds great promise for the treatment of long filaments. A second interesting finding concerns the significant increase in Zn in the printed samples as a result of an increase in the concentration of dissolved PLA in the coating dispersion, as appears in Figure 6. This alludes to an interaction between the PLA and the ZnO NPs, which is supported by the fact that the addition of PLA in the ZnO NP solution increases the dispersion’s stability and prevents the aggregation of the NPs (Appendix A). The recurrence of the Zn concentration in the printed samples, despite being printed from different sections of the filament (thanks to the layer-by-layer printing of multiple samples), emphasizes the homogeneity of the additive throughout the printed layers.

Obtaining long-lasting antibacterial surfaces based on metal oxide NPs requires minimal leaching of the inorganic material. Hence, we examined the leaching of Zn(II) ions over time from the coated printed samples placed in a PBS solution (120 rpm at 37 °C) for nine weeks. Less than 5 μg of Zn(II) ions leaching out of every gram of the printed samples was detected after nine weeks (a less than 2.5% loss), implying the very high durability of the additive NPs within the polymer.

Tensile strength testing was carried out to ensure that the mechanical properties of the printed samples did not change due to the coating process. Hence, the coated and new filaments were tested after being printed as dogbones in two orientations, XY and Z (Appendix A). Treated filaments were coated using 10 and 20 mg/mL of dissolved PLA with up to 15 mg/mL ZnO NPs and without any additives. XY samples were printed with the large face of the structure on the heated bed, thus enabling testing of the PLA strands melted one onto the other at 90° layers. Z samples were printed upwards, applying the pressure of the Ingstrom machine during the measurement against the layers’ connection. It was found that the Young’s modulus, *E*, and the yield strength of the printed XY- and Z-oriented structures are only slightly different (Table 1). Furthermore, the addition of the additives, i.e., PLA and ZnO NPs, did not affect the mechanical properties of the samples in both directions. Table 1 presents the average results of the samples made from all the mentioned filaments, both pristine and treated. It is important to note that only rarely did the XY-oriented samples break. The XY-oriented samples broke infrequently; most survived the maximum tensile stress of the machine. Hence, the reported values represent the minimum ultimate strength and maximum elongation of the XY-oriented samples. Since only the outer layer of the filament is altered, known systems and techniques for strengthening 3D-printed objects such as fiber impregnation or multi-material printing can theoretically be used to further tailor the filament’s mechanical properties for specific applications [15,16].

ZnO NPs are often used for biological applications due to their effective antibacterial properties and structure-dependent physicochemical properties [41]. Therefore, we conducted antibacterial tests to assess the effectiveness of the coating on the printed samples (Figure 7). The test involved measuring the optical density of the solution containing the bacteria after being exposed for 4 h to the printed samples, and an additional 20 h of incubation inside a Lysogeny broth (LB). Hence, the lower the *O.D.*, the higher the antibacterial activity (see Section 2.3.4 for more details).

Three important trends can be seen in Figure 7: (i) As expected, the more ZnO NPs present in the coating process, the stronger the antibacterial effect. (ii) Unexpectedly, the Gram-negative *E. coli* showed a stronger viability than the Gram-positive *S. aureus*, requiring more ZnO NPs to prohibit its extermination. This stands against numerous studies and theories regarding the weaker resistance of Gram-negative bacteria to metal oxide NPs in general, and the activity of ZnO NPs against *E. coli* in comparison to *S. aureus,* in particular [39,42]. (iii) A stronger or equal antibacterial activity against *E. coli* was found for all ZnO NP concentrations higher than 5 mg/mL when 10 mg/mL of PLA was used. We recall that the amount of ZnO NPs increases with that of PLA in the coating solution (Figure 6). For this reason, we expected to find a greater eradication of bacteria on the samples made with 20 mg/mL of PLA (as seen for *S. aureus*) against both bacteria. 

The two surprising trends can be explained through possible compatibilities between *E. coli* and FDM-printed PLA samples. We hypothesize that the cracks and pores on the samples’ surface are compatible with the growth pattern of *E. coli*, thus acting as an optimal breeding site for the pathogen. A coating made using a high concentration of PLA could have resulted in a greater abundance of defects in the filament and print or a looser layer, providing additional optimal sites for bacteria growth. A rather similar effect in different nanoscale topographies on the growth patterns of bacteria, including *E. coli*, was reported by Hsu et al. and by Chinnaraj et al., supporting our reasoning for this phenomenon [43,44]. This claim is supported by the higher *O.D.* of bacteria found on the samples with 20 and 10 mg/mL of dissolved PLA where no ZnO NPs were suspended (1.7 ± 0.3 and 1.5 ± 0.1 a.u. after 20 h of incubation, respectively). However, an in-depth comparison of the growth speed of the different bacteria colonies on FDM-printed samples is beyond the scope of this research. 

In summary, ZnO NPs dispersed in a hydrophobic phase for a long period of time were used to homogeneously coat a PLA filament. The embedded ZnO NPs were stable in the printed objects for nine weeks under physiological conditions and showed high antibacterial activity against both Gram-negative and Gram-positive bacteria, indicating their potential for use in biological implants and prostheses where long-term stability and anti-infection are sought. 

### 3.3. The Cip Coating

To further test for other capabilities of CMDF, we loaded small organic molecules and followed their release rates and kinetics. The model material was Ciprofloxacin (Cip), known for its broad spectrum of antibacterial activity, possible anticancer activity, and relative ease of detection [45,46]. Cip is a proper candidate for such tests, as it has already shown in vivo promise as a medical additive for preprinting surgical implants in rabbits [25].

The release profile of the Cip was followed through UV-light absorption measurements for an incubation time of up to 35 days in PBS (Figure 8) for printed samples coated in a solution containing 20 mg/mL PLA and 5 to 50 mg/mL of dissolved Cip. 

It was found that the release profile closely matches the Ritger–Peppas theoretical and empirical model for material diffusion out of a reservoir-type device (Appendix A) for the mid-/long-term administration of medication through insertion. In that model, the diffusing materials are non-covalently embedded within a polymeric structure and are released under a rate-controlling polymeric membrane through pores [26]. A faster rate and a longer time-frame until saturation were observed to be linked to the concentration of the antibiotics, where faster rates were reached for higher concentrations. It should be noted that we also carried out antibacterial tests and found the full eradication of both bacteria on the printed samples containing Cip. As the applied method consisted of prolonged direct exposure of a highly concentrated bacteria suspension in small volumes to the surface and a clear direct Cip release was found, the result was as expected.

## 4. Conclusions

Continuous Material Deposition on Filaments (CMDF) is a new approach to the in situ continuous coating of FDM filaments for use in 3D printing. This is based on passing the filament through a solution in which the coating material is dissolved or dispersed together with the same polymer as the filament. This approach shows promising results in forming thin homogeneous layers on a polymeric filament for FDM printing, enabling a variety of active materials to be easily and effectively embedded within 3D structures printed by FDM technology. As models, both organic molecules and metal oxide NPs were successfully coated in a thin layer onto the filament and were preserved during the printing of the 3D structures. The functionality of the embedded material was demonstrated by prolonged antibacterial activity due to coating the filament with ZnO NPs and the controlled release of the antibiotic Cip. This allows for persistent and localized benefits against infection, a crucial property for medical implants. Importantly, the method does not alter the mechanical properties of the printed structures. Considering the promising results, the simplicity of the method, and the construction of the fabrication setup, we expect that CMDF can be adapted to rapidly impart a functional property onto commercially available filaments with proper tailoring of the composition of the coating materials.

## Figures and Tables

**Figure 1 polymers-16-02904-f001:**
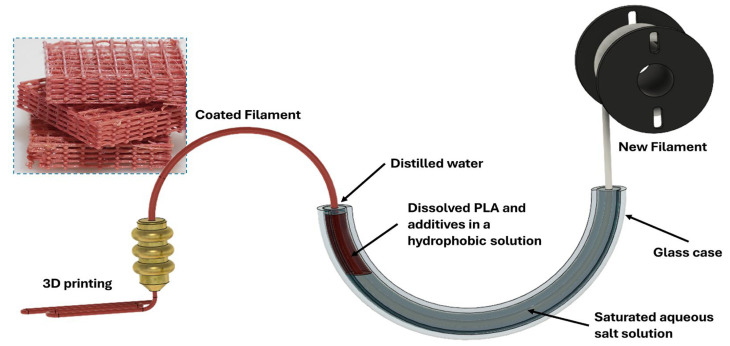
A schematic representation of the Continuous Material Deposition on Filaments treatment (CMDF). A new strand of filament moves through a glass tube through three different liquid phases; a salt-saturated aqueous phase, a Dichloromethane/Tetrahydrofuran (DCM/THF) hydrophobic phase (where the coating occurs), and a thin layer of distilled water, before being dried and used in a 3D printer.

**Figure 2 polymers-16-02904-f002:**
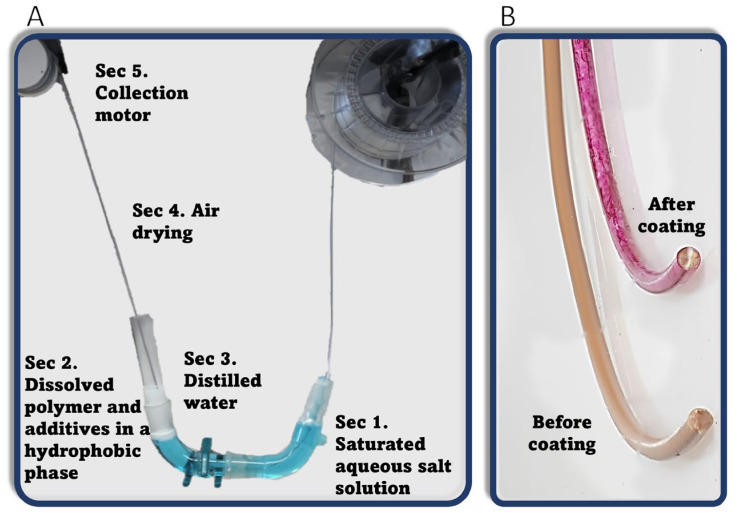
(**A**) Photo of CMDF. (**B**) Treated filament before and after embedding Rhodamine B.

**Figure 3 polymers-16-02904-f003:**
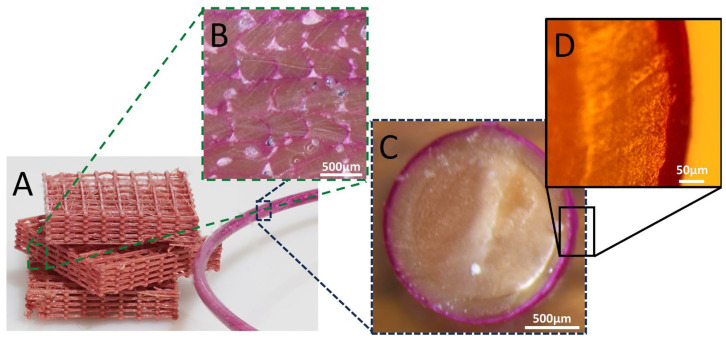
(**A**) The Rhodamine B-coated filament and printed sample with cross-section optical microscopy images of (**B**) The printed sample and (**C**,**D**) The filament.

**Figure 4 polymers-16-02904-f004:**
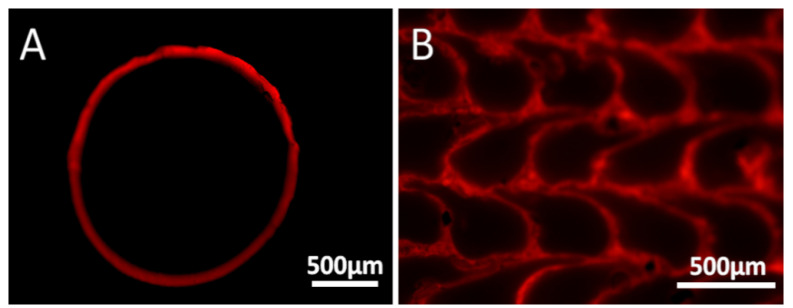
Cross-section imaging by fluorescence microscopy of the Rhodamine B-coated filament before (**A**) and after (**B**) printing.

**Figure 5 polymers-16-02904-f005:**
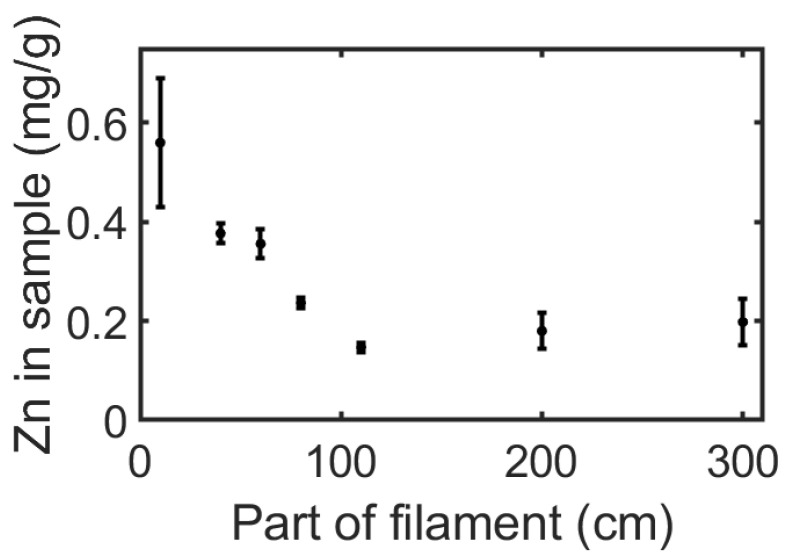
The amount of Zn in a coated filament at different locations along the filament. The coating dispersion contains 20 mg/mL of polylactide acid (PLA) and 15 mg/mL of ZnO nanoparticles (NPs). Each point on the graph refers to 20 cm of filament centered of the cut portion of the filament.

**Figure 6 polymers-16-02904-f006:**
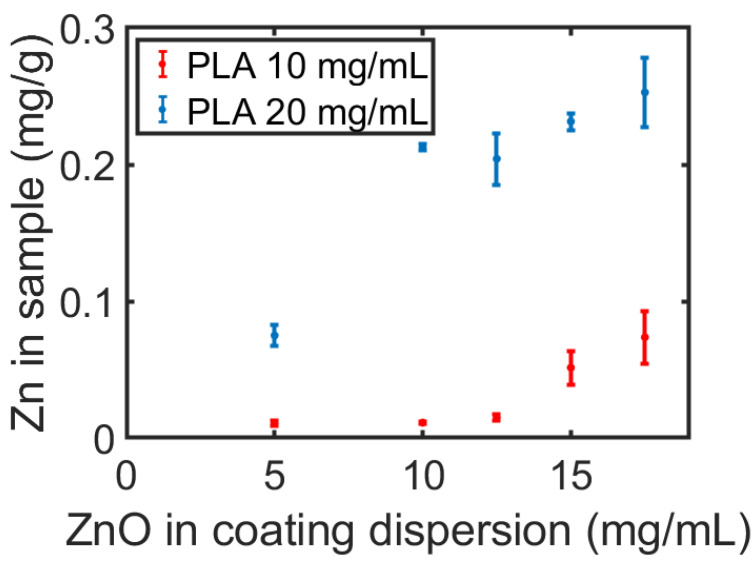
The amount of zinc in a sample as a function of the concentration of ZnO NPs in the dispersion containing 10 or 20 mg/mL of PLA.

**Figure 7 polymers-16-02904-f007:**
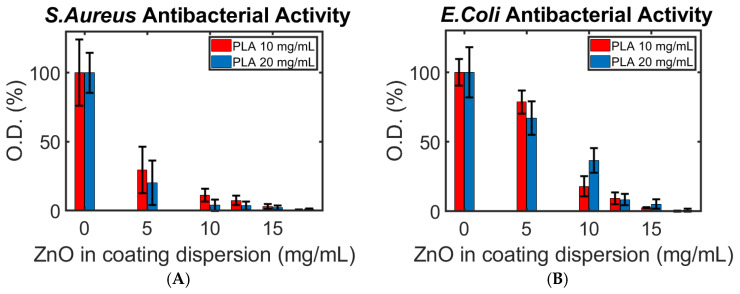
The antibacterial activity of (**A**) *S. aureus* and (**B**) *E. coli* at different concentrations of PLA and ZnO NPs. The concentrations refer to the coating solutions tested. The bacteria were exposed directly to the printed samples.

**Figure 8 polymers-16-02904-f008:**
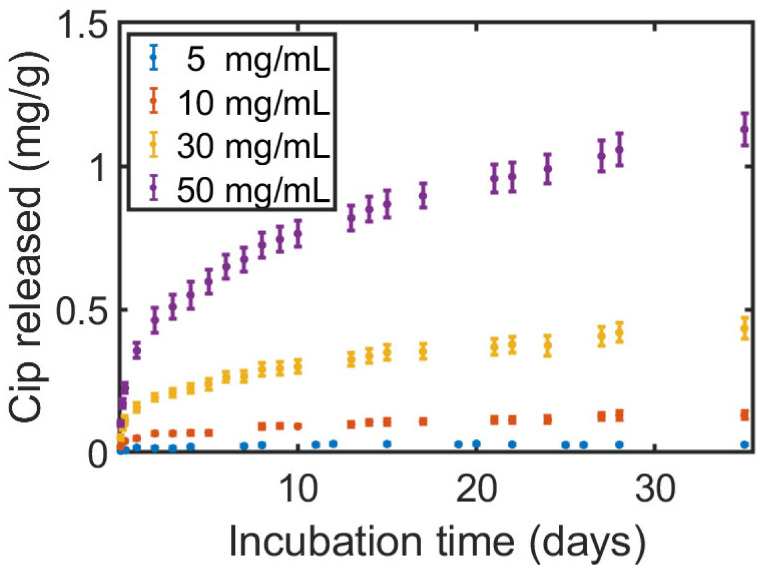
The accumulative release–mass ratio of Ciprofloxacin (Cip) released into the PBS solution from samples printed with a filament passed through a coating solution of 20 mg/mL PLA and 5–50 mg/mL of dissolved Cip.

**Table 1 polymers-16-02904-t001:** The average mechanical properties of the samples printed in two orientations, XY and Z. The average is calculated for both treated and untreated filaments, as no noticeable difference was found between the two.

	XY Orientation	Z Orientation
E (MPa)	530 ± 40	510 ± 10
Yield strength (MPa)	22 ± 3	19 ± 1
Ultimate strength (MPa)	22 ± 2 *	19.6 ± 0.7
Max elongation (%)	5.6 ± 0.4 *	7.1 ± 0.8

* The minimal mean result at the breaking point. Most samples are above this value.

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
