# Peer review of "Continuous Material Deposition on Filaments in Fused Deposition Modeling"

_polymers, 2024, doi:10.3390/polym16202904_

Round 1

Reviewer 1 Report

Comments and Suggestions for Authors

Reviewer comments 

Manuscript ID: Polymers-3180721

Title: Continuous Material Deposition on Filaments in Fused Deposition Modeling

Journal: Polymers

The aims of the paper are to explore the integration of active materials into 3D-printed objects using a three-phase coating system and to demonstrate the potential applications of this technique in fields such as medicine and agriculture. Specifically, the study aims to fabricate 3D objects through fused deposition modeling (FDM) with embedded functional materials using the Continuous Material Deposition on Filaments (CMDF) method. The paper also aims to assess the effectiveness of this method by testing different materials (Rhodamine B, ZnO nanoparticles, and Ciprofloxacin) dissolved or dispersed in a PLA-containing solution, and to evaluate whether the properties of the coating materials are preserved throughout the coating and printing processes.

The paper falls within the scope of the journal; however, I believe it requires some enhancement. I propose the following revisions to improve the paper.

1-      The aims of the paper should be clearly described in the abstract.

2-      The printing parameters should be presented in the manuscript. Additionally, it should be clarified whether there are any changes in the printing parameters when integrating active materials into 3D-printed objects.

3-      The authors claim that the mechanical properties of the coated materials do not change with the addition of active materials based on Young's modulus. However, information about other mechanical properties such as yield strength, ultimate strength, and properties in different directions should also be included. Furthermore, the bonding between layers with the addition of coating material should be evaluated to confirm that these mechanical properties remain unchanged.

4-      Line 188: Please verify the reference to Figure S1, as it does not appear in the manuscript.

5-      The authors claim in line 189 that they are “…testing the mechanical properties of the mesh and the adhesion between layers.” More information on how the adhesion between layers is tested should be provided, along with results that verify the influence of active materials on the adhesion between layers of the coating materials.

6-      Lines 264-267: reformulate this sentence

7-      Typos and grammar problems need to be corrected properly, authors should carefully check through the manuscript before submitting a revision. 

The reviewer recommends that the author do major revision to the manuscript.

Comments on the Quality of English Language

The reviewer would recommend that the authors proofread the article thoroughly for typos and grammatical errors.

Author Response

Reviewer 1

Dear reviewer,

Thank you for your comments. Below are our replies to the specific comments. The changes in the revised manuscript were highlighted.

  • The aims of the paper should be clearly described in the abstract.

The abstract has been revised to better describe the goals and intentions of our design, specifically focusing more on our work as model cases for a coating system and our endeavors to prove its usefulness.

  • The printing parameters should be presented in the manuscript. Additionally, it should be clarified whether there are any changes in the printing parameters when integrating active materials into 3D-printed objects.

This information was already presented in the Supplementary Section of the original manuscript (Table S1) and was referred to in the main text (section 2.3.3).

  • The authors claim that the mechanical properties of the coated materials do not change with the addition of active materials based on Young's modulus. However, information about other mechanical properties such as yield strength, ultimate strength, and properties in different directions should also be included. Furthermore, the bonding between layers with the addition of coating material should be evaluated to confirm that these mechanical properties remain unchanged.

To address your concerns we have added the results of maximum elongation at break, yield strength, and ultimate strength in the revised manuscript. A new table (Table 1) presenting the mechanical properties was added to the manuscript.

  • Line 188: Please verify the reference to Figure S1, as it does not appear in the manuscript.

The reference to Figure S1 was already in the original manuscript (section 2.3.6).

  • The authors claim in line 189 that they are “…testing the mechanical properties of the mesh and the adhesion between layers.” More information on how the adhesion between layers is tested should be provided, along with results that verify the influence of active materials on the adhesion between layers of the coating materials.

The adhesion between the layers was estimated through the mechanical properties of objects printed in different directions. The bonding between the layers was indicated by measuring dogbone shapes in the Z direction. The Z-orientation printed samples appear in Figure S1. We find that the Young modulus and the yield strength for the two orientations are similar. 

  • Lines 264-267: reformulate this sentence

As per your suggestion, the sentence was rewritten and restructured into two shorter sentences.

  • Typos and grammar problems need to be corrected properly, authors should carefully check through the manuscript before submitting a revision.

We reread the paper and corrected every typo or grammatical error.

Reviewer 2 Report

Comments and Suggestions for Authors

Continuous Material Deposition on Filaments in Fused Deposition Modeling

This article explores the development of a new method called Continuous Material Deposition on Filaments (CMDF) for enhancing 3D-printed structures using Fused Deposition Modeling (FDM). The CMDF technique allows for the uniform coating of polymer filaments with active materials, such as Rhodamine B, ZnO nanoparticles, and Ciprofloxacin. These materials maintain their functionality after printing, enabling applications like antimicrobial surfaces and controlled drug release. The study demonstrates that the mechanical properties of the printed objects remain unaffected by the coating process. CMDF offers a simple, effective way to incorporate functional materials into FDM-printed objects, opening possibilities for medical, agricultural, and other fields.

Comments:

1.     The abstract needs major revisions. The current version lacks a clear presentation of innovation, the purpose of research, quantitative and qualitative results, and achievements. Therefore, the abstract should be rewritten and all mentioned items should be added to it.

2.     What is the significance of incorporating active materials like ZnO nanoparticles into 3D-printed structures using fused deposition modeling (FDM)?

3.     How does the Continuous Material Deposition on Filaments (CMDF) technique ensure uniformity in coating filaments for FDM printing?

4.     What were the mechanical effects of embedding ZnO nanoparticles on the PLA filament as observed during tensile testing?

5.     How does the preprinting treatment of filaments differ from post-printing treatments in terms of material uniformity and release rates?

6.     What challenges does the CMDF method address compared to other filament coating techniques like continuous bath coating?

7.     Use the following papers to deepen the introduction and discussion. Advancing sustainable shape memory polymers through 4D printing of polylactic acid-polybutylene adipate terephthalate blends. Various FDM mechanisms used in the fabrication of continuous-fiber reinforced composites: a review.

8.     How did the addition of Ciprofloxacin to the 3D-printed structures influence the controlled release of antibiotics over time?

9.     What role did Rhodamine B play in the experimental demonstration of the CMDF technique, and how was its effectiveness evaluated?

10.  What are the potential biomedical applications of CMDF-coated filaments, particularly in antimicrobial surfaces for medical implants?

11.  What were the observed effects of ZnO nanoparticle coatings on the antibacterial properties of 3D-printed structures against E. Coli and S. Aureus?

Comments on the Quality of English Language

See the comments.

Author Response

Reviewer 2

Dear reviewer,

Thank you for your comments. Below are our replies to the specific comments. The changes in the revised manuscript were highlighted.

  1. The abstract needs major revisions. The current version lacks a clear presentation of innovation, the purpose of research, quantitative and qualitative results, and achievements. Therefore, the abstract should be rewritten and all mentioned items should be added to it.

The abstract has been revised to better describe the goals and the results, as suggested.

  1. What is the significance of incorporating active materials like ZnO nanoparticles into 3D-printed structures using fused deposition modeling (FDM)?

In general, incorporating active materials within 3D printed structures is important in a variety of fields. For example, obtaining devices with slow release of functional materials, imparting antibacterial activity to the objects and more. ZnO provides antibacterial activity and is therefore an examples of introducing nanoparticles with specific activity to the printed object. Usually, to incorporate materials within a printed strcutre requires developing new compositions for each material. Here we propose an alternative approach, which is simpler and highly generic. This text was added in the Conclusions section. 

  1. How does the Continuous Material Deposition on Filaments (CMDF) technique ensure uniformity in coating filaments for FDM printing?

In principle, an inherent property of the proposed method is the uniformity of the coating of the filament under the condition that it is fully wetted by the coating solution. Indeed, the uniformity of the coating was verified by microscopy as shown in Figures 3 and 4 and by following the concentration of Zn at different parts of the coated filament and printed objects (Figures 5 and 6). A few sentences were added throughout the revised paper to further clarify this topic.

  1. What were the mechanical effects of embedding ZnO nanoparticles on the PLA filament as observed during tensile testing?

The mechanical effects of embedding ZnO NPs were negligible for tensile testing. A table showing the full results and a further explanation of the topic was added to the results (Table 1 in the revised version).

  1. How does the preprinting treatment of filaments differ from post-printing treatments in terms of material uniformity and release rates?

The difference between preprinting treatment of filaments from post-printing treatments of printed objects is mainly in the accessability of the active material throughout the printed object. Typical post-treatment includes dip- and spray coating, and electroless plating as explained in the introduction. This enables embedding active materials only at the outer layer of the whole object. Such studies were discussed in the Introduction section.

  1. What challenges does the CMDF method address compared to other filament coating techniques like continuous bath coating?

As discussed in the introduction, several problems are associated when applying continuous bath coating such as damaging the coating while forming it as well as the formation of non-uniform coatings. In our system, since we have an aqueous layer, we obtain uniform coatings while avoiding exposure to harmful solvents, such as chloroform. The limitation of the continuous bath coating were discussed in the Introduction section and in order to clarifiy the reason behind the aqueous layer, several sentences were added.

  1. Use the following papers to deepen the introduction and discussion. Advancing sustainable shape memory polymers through 4D printing of polylactic acid-polybutylene adipate terephthalate blends.Various FDM mechanisms used in the fabrication of continuous-fiber reinforced composites: a review.

Thanks for the references. Both papers are interesting and were added to the Introduction section to explain how treatments can be carried out during the printing process, and also in the Discussion and Results section to further explain that a combination of multiple treatment systems can be used to obtain functional prints.

  1. How did the addition of Ciprofloxacin to the 3D-printed structures influence the controlled release of antibiotics over time?

The addition of Ciprofloxacin allowed its release over time as shown in Figures 8 and S6. Evidently, we cannot compare the results to other cases werehe Cip is introduced into the filament itself as such filaments are not available. An explanation of the topic was added to the Results and Discussion section and appears in depth in the Supporting Information.

  1. What role did Rhodamine B play in the experimental demonstration of the CMDF technique, and how was its effectiveness evaluated?

Rhodamine B was applied as an easy-to-track material for the microscopical, fluorescence, and photographic characterization of the process, as can easily be seen in the pictures provided in Figures 3 and 4. The reason behind choosing Rhodamine B was rewritten to better explain our reasoning. We did not see any change in the absorption or emission of Rhodamine B implying that it was not affected by the adsorption process.

  1. What are the potential biomedical applications of CMDF-coated filaments, particularly in antimicrobial surfaces for medical implants?

Antimicrobial surfaces are well sought after as they prevent infections or allow sustained drug delivery. The point was restated in a few places in the revision as well as the preliminary choice to use medical purposes as a model to show the usefulness of CMDF.

  1. What were the observed effects of ZnO nanoparticle coatings on the antibacterial properties of 3D-printed structures against E. Coli and S. Aureus?

Figure 7 shows the observed effects of ZnO nanoparticle coatings on the antibacterial properties against the two tested bacteria. In order to better emphasize that the tests were conducted on coated samples that were treated preprinting, the description of the figure was expanded and the appropriate text referring to it was revised in section 2.3.4.

Reviewer 3 Report

Comments and Suggestions for Authors

I am in favor of publishing this manuscript containing interesting data and relevant analysis. I would suggest the authors to consider following recommendations carefully and resubmit their manuscript after through revision (please include the new information in the text, for each):

 1. “the filament is immersed within a half-circular tube containing an aqueous salt solution, with a PLA saturated dichloromethane (DCM)/tetrahydrofuran (THF) (1:1 v/v) solution on top of the aqueous solution, covered at the end with a distilled water layer.” - Please mention the volume of each layer of liquid used.

 2. Section 2.1: Were the chemicals further purified or used as received?

 3. “PLA filaments were purchased from Esun (Chi Minh, Vietnam) and PRUSA (Prague, Czech Republic)” – Was there any difference in yarn properties which were collected from two different sources?

 3b. Why did the authors use two different sources?

 3c. Please clearly mention the application (dissolved/filament for passing) of the filament obtained from each source.  

 4. What was the diameter of the monofilaments mentioned in section 2.1?

 5. “Clear PLA filament is added and dissolved under continuous stirring.” – Please mention the weight of the filament used. What was the end concentration of the solution?

 6. “The semicircle glass tube (shown schematically in Figure 1 and a photo in Figure 2) of the coating system was constructed with glassware…” – Please mention the length of the tube.

 7. “the coated filament goes through a thin layer of distilled water (Figure 2A-Sec. 3) which cleans the filament” -  Distilled water would be contaminated fast during the process. How was the liquid inside the tube kept clean and uniform during the coating?

 8. “Tensile strength testing was carried out to ensure that the mechanical properties of the filaments did not change during the coating process.” – Please provide with a data table in the main text summarizing the tensile properties.

 9. Line 296 - 301: Please provide appropriate references.   

 10. The English language of this manuscript needs some careful review.

Author Response

Reviewer 3

Dear reviewer,

Thank you for your comments. Below are our replies to the specific comments. The changes in the revised manuscript were highlighted.

  1. “the filament is immersed within a half-circular tube containing an aqueous salt solution, with a PLA saturated dichloromethane (DCM)/tetrahydrofuran (THF) (1:1 v/v) solution on top of the aqueous solution, covered at the end with a distilled water layer.” - Please mention the volume of each layer of liquid used.

As requested, the volumes were added to section 2.3.2 (Coating Setup section) as suggested.

  1. Section 2.1: Were the chemicals further purified or used as received?

The chemicals were used as provided. This information was added under the materials (section 2.1).

  1. “PLA filaments were purchased from Esun (Chi Minh, Vietnam) and PRUSA (Prague, Czech Republic)” – Was there any difference in yarn properties which were collected from two different sources?

            3b. Why did the authors use two different sources?

            3c. Please clearly mention the application (dissolved/filament for passing) of the filament obtained from each source.  

The filaments were used interchangeably as no differences were easily observed for our needs. The main reason behind the usage of multiple materials was the variety of colors. Namely, when working with Rhodamine B, we found that the best pictures were obtained when coating a light brown filament bought from Esun rather than the grey filament bought from PRUSA. This fact was added to the paper.

  1. What was the diameter of the monofilaments mentioned in section 2.1?

The diameter (1.75 mm) was added to the appropriate location. This diameter was chosen to present the most common size used for FDM printing.

  1. “Clear PLA filament is added and dissolved under continuous stirring.” – Please mention the weight of the filament used. What was the end concentration of the solution?

The concentrations and weight of filament dissolved in the coating solution are mentioned in every figure where different concentrations were used. The maximum concentration dissolved was restated in section 2.3.1

  1. “The semicircle glass tube (shown schematically in Figure 1 and a photo in Figure 2) of the coating system was constructed with glassware…” – Please mention the length of the tube.

The length of the bath (35 cm) was added to section 2.3.2

  1. “the coated filament goes through a thin layer of distilled water (Figure 2A-Sec. 3) which cleans the filament” - Distilled water would be contaminated fast during the process. How was the liquid inside the tube kept clean and uniform during the coating?

The Introduction section was revised to better address this point. The solution layer cleans the filament from residues of the previous layer as hydrophobic organic materials are expected to remain in the DCM/THF layer. As the sentence states, the main purpose of the solution layer is to prevent the organic phase from evaporating. The importance of using distilled water as one aqueous layer and saturated salt solution as the other is the different densities of the solution. This causes one layer to lie above the hydrophobic phase while the other layer is below it.

  1. “Tensile strength testing was carried out to ensure that the mechanical properties of the filaments did not change during the coating process.” – Please provide with a data table in the main text summarizing the tensile properties.

A table with all the relevant mechanical properties was added to the revised manuscript and the majority of the discussion around mechanical properties was rewritten to highlight the tensile properties.

  1. Line 296 - 301: Please provide appropriate references. 

Lines 296-301 in the original manuscript present a hypothesis trying to explain unexpected results where E. Coli proliferated better on samples printed with 20 mg/mL rather than 10 mg/mL of PLA in the coating solution. We hypothesized that the concentration of the PLA has a strong effect on the topography of the printed object. Two previous papers reported a somewhat similar effect where proliferation was related to the topography of the surface. Studying the effect of the concentration of PLA in the deposition solution on the proliferation of bacteria is beyond this study and therefore, we raised the hypothesis and associated it with previous reports.

Reviewer 4 Report

Comments and Suggestions for Authors

The manuscript entitled "Continuous Material Deposition on Filaments in Fused Deposition Modeling" is focused on embedding of the active functional materials in printed 3D objects. The presented approach for the preprinting treatment of fused deposition modeling filaments is rather a form of technological innovation than scientific discovery. Nevertheless, the work seems worth publishing due to the possible practical application of the presented method.

 The manuscript can be improved according to the following suggestions:

1. The section "introduction" is a little long-winded. The authors describe in detail even the basic technical aspects of preprinting and post-printing treatment. However, there is a lack of application examples and in particular justification for the selection of active materials: Rhodamine B, ZnO nanoparticles and ciprofloxacin. The attempts to justify the use of the chosen antibiotic appear only at the end of the results section.

2) The "Results and Discussion" section contains a few general sentences that fit better into the introduction, e.g. "Nanomaterials are particles with at least one dimension below 100 nm. The most studied nanomaterials are NPs, benefiting medicine, energy harvesting and storage, and sensing. The inclusion of nanomaterials within 3D objects can bring unique advantages. Metal oxide NPs…", etc.

3) The letters (B, C, D) in the individual panels in Figure 3 are barely visible on the A4 printout. The font seems to be incorrectly embedded using the graphics program (white outlines are visible around the letters).

4) In Figure 8 it is better to write 5 [mg/ml] rather than 05 (zero is unnecessary, because it may suggest the number 0.5). The release profile of the ciprofloxacin for 5 mg/ml is difficult to analyze in such a graph (error bars are also invisible for this amount of dissolved Cip).

Comments on the Quality of English Language

The language of the manuscript is generally understandable and the work does not contain many glaring linguistic errors.

Author Response

Reviewer 4
Dear reviewer,
Thank you for your comments. Below are our replies to the specific comments. The changes in the revised manuscript were highlighted.
1. The section "introduction" is a little long-winded. The authors describe in detail even the basic technical aspects of preprinting and post-printing treatment. However, there is a lack of application examples and in particular justification for the selection of active materials: Rhodamine B, ZnO nanoparticles and ciprofloxacin. The attempts to justify the use of the chosen antibiotic appear only at the end of the results section.
Thank you for your useful comment. As you mentioned in your opening statement, this research is focused on the conception of a new and improved system rather than specific research with a singular goal or field. In order to highlight the necessity of our approach, we find it crucial to introduce and review the possible alternatives of treatments for FDM printing. This resulted in a somewhat lengthy Introduction. Justification of the chosen active materials has been added to the Introduction section. 
2) The "Results and Discussion" section contains a few general sentences that fit better into the introduction, e.g. "Nanomaterials are particles with at least one dimension below 100 nm. The most studied nanomaterials are NPs, benefiting medicine, energy harvesting and storage, and sensing. The inclusion of nanomaterials within 3D objects can bring unique advantages. Metal oxide NPs…", etc.
For a similar reason as mentioned above and to allow a distinct flow of the text, the explanations regarding the exemplary additives were kept also in the Results and Discussion section. In practice, every material or measurement we chose to test, could have been replaced by another without changing the goal or conclusion of the paper as the properties of the coating system would have been kept all the same. That being said, we have extended the initial presentation of the materials and refer the reader to keep reading to better understand the reasoning behind them.
3) The letters (B, C, D) in the individual panels in Figure 3 are barely visible on the A4 printout. The font seems to be incorrectly embedded using the graphics program (white outlines are visible around the letters).
Thank you for drawing our attention to this issue. The figure was fixed to allow a better visibility of the letters.
4) In Figure 8 it is better to write 5 [mg/ml] rather than 05 (zero is unnecessary, because it may suggest the number 0.5). The release profile of the ciprofloxacin for 5 mg/ml is difficult to analyze in such a graph (error bars are also invisible for this amount of dissolved Cip).
The graph has been updated to exclude the unnecessary zero. However, the release profile of 5 mg/ml was not changed. While we admit that it is nearly impossible to draw clear conclusions from this data set, it does present the limitations of the system for the inclusion and release of Cip, therefore we see the merit of providing this data to the reader.

Round 2

Reviewer 1 Report

Comments and Suggestions for Authors

After reviewing the revisions, the paper now meets the necessary standards for publication. I recommend accepting the manuscript

Comments on the Quality of English Language

ok

Reviewer 3 Report

Comments and Suggestions for Authors

Acceptance is recommended.